# Fully Human Monoclonal Antibodies Effectively Neutralizing Botulinum Neurotoxin Serotype B

**DOI:** 10.3390/toxins12050302

**Published:** 2020-05-07

**Authors:** Takuhiro Matsumura, Sho Amatsu, Ryo Misaki, Masahiro Yutani, Anariwa Du, Tomoko Kohda, Kazuhito Fujiyama, Kazuyoshi Ikuta, Yukako Fujinaga

**Affiliations:** 1Department of Bacteriology, Graduate School of Medical Sciences, Kanazawa University, Takara-machi, Kanazawa, Ishikawa 920-8640, Japan; t-matsu@med.kanazawa-u.ac.jp (T.M.); amatsu@med.kanazawa-u.ac.jp (S.A.); yutanim@med.kanazawa-u.ac.jp (M.Y.); 2Applied Microbiology Laboratory, International Center for Biotechnology, Osaka University, Yamadaoka, Suita, Osaka 565-0871, Japan; misaki@icb.osaka-u.ac.jp (R.M.); fujiyama@icb.osaka-u.ac.jp (K.F.); 3Department of Virology, Center for Infectious Disease Control, Research Institute for Microbial Diseases, Osaka University, Yamadaoka, Suita, Osaka 565-0871, Japan; anariwa3100@163.com (A.D.); ikuta@iph.osaka.jp (K.I.); 4Department of Veterinary Sciences, School of Life and Environmental Sciences, Osaka Prefecture University, Rinkuouraikita, Izumisano, Osaka 598-8531, Japan; kohda@vet.osakafu-u.ac.jp; 5The Japan Science and Technology Agency/Japan International Cooperation Agency, Science and Technology Research Partnership for Sustainable Development, Tokyo 102-0076, Japan

**Keywords:** *Clostridium botulinum*, neurotoxin, botulism, fully human monoclonal antibody, SPYMEG, therapeutic effect, preventive effect

## Abstract

Botulinum neurotoxin (BoNT) is the most potent natural toxin known. Of the seven BoNT serotypes (A to G), types A, B, E, and F cause human botulism. Treatment of human botulism requires the development of effective toxin-neutralizing antibodies without side effects such as serum sickness and anaphylaxis. In this study, we generated fully human monoclonal antibodies (HuMAbs) against serotype B BoNT (BoNT/B1) using a murine–human chimera fusion partner cell line named SPYMEG. Of these HuMAbs, M2, which specifically binds to the light chain of BoNT/B1, showed neutralization activity in a mouse bioassay (approximately 10 i.p. LD_50_/100 µg of antibody), and M4, which binds to the C-terminal of heavy chain, showed partial protection. The combination of two HuMAbs, M2 (1.25 µg) and M4 (1.25 µg), was able to completely neutralize BoNT/B1 (80 i.p. LD_50_) with a potency greater than 80 i.p. LD_50_/2.5 µg of antibodies, and was effective both prophylactically and therapeutically in the mouse model of botulism. Moreover, this combination showed broad neutralization activity against three type B subtypes, namely BoNT/B1, BoNT/B2, and BoNT/B6. These data demonstrate that the combination of M2 and M4 is promising in terms of a foundation for new human therapeutics for BoNT/B intoxication.

## 1. Introduction

Botulinum neurotoxins (BoNTs) produced by the anaerobic bacterium *Clostridium botulinum* and related species cause botulism, a neuroparalytic disease with high mortality [1,2]. BoNTs have been classified as category A agents by the Centers of Disease Control and Prevention (CDC) and are listed among the six agents at highest risk of being used as bioweapons [3]. Originally, seven serotypes, designated A to G, have been identified, and four of these, namely A, B, E, and F, cause human botulism [2]. Additionally, BoNT/DC, which is considered a mosaic toxin between BoNT/D and BoNT/C, has been reported [4,5]. Recently, BoNT/H was reported [6,7], and subsequent studies have described that this toxin is a hybrid-toxin of BoNT/A1 and BoNT/F5 [8,9,10] and that its light chain and N-terminal of its heavy chain are immunologically unique [11,12]. More recently, the novel serotype BoNT/X [13] and BoNT-like toxin, BoNT/Wo [14] and BoNT/En (BoNT/J) [15,16], were also reported [17].

Each BoNT is synthesized as a single polypeptide chain (150 kDa) that is proteolytically activated by cleavage into a light chain (L chain, 50 kDa) and a heavy chain (H chain, 100 kDa), which are linked by a disulfide bond [1]. The L chain acts as a zinc metalloprotease. The H chain consists of two functionally distinct regions: the C-terminal, or receptor-binding, domain (H_C_), and the N-terminal, or translocation, domain (H_N_).

Food-borne and infant botulism are the primary forms of human botulism [18], and are caused by intestinal absorption of BoNT. BoNT in the gastrointestinal lumen crosses the intestinal barrier [19], enters the blood stream, and reaches the neuromuscular junction. There, BoNT binds via H_C_ to the receptors present on presynaptic nerve terminals. BoNT/A, BoNT/D, BoNT/E, and BoNT/F bind to synaptic vesicle protein 2 (SV2) and polysialogangliosides [20,21,22], whereas BoNT/B and BoNT/G bind to synaptotagmin and polysialogangliosides [23,24]; all serotypes subsequently enter neuronal cells by endocytosis. In the acidified synaptic vesicles, H_N_ induces translocation of the L chain into the cytosol [25,26,27]. The metalloprotease domain of the L chain cleaves the soluble *N*-ethylmaleimide–sensitive fusion protein attachment protein receptors (SNAREs) required for synaptic vesicle fusion. The synaptosomal associated protein 25-kDa (SNAP-25) is the target of BoNT/A, BoNT/C, and BoNT/E; synaptobrevin (also known as vesicle-associated membrane protein, VAMP) is the target of BoNT/B, BoNT/D, BoNT/F, and BoNT/G; and syntaxin is the target of BoNT/C [1,2]. Cleavage of SNARE inhibits release of the neurotransmitter acetylcholine and leads to paralysis [1,2].

Immunotherapy is the most effective treatment for BoNT intoxication. Equine immune serum formulations are used in cases of human botulism: Botulism-Antitoxin Behring (Novartis Vaccines and Diagnostics GmbH and Co. KG) is used for treatment of types A, B, and E, while Botulism Antitoxin Heptavalent (BAT, Cangene Corporation) is used for treatment of types A, B, C, D, E, F, and G. However, because both drugs use heterologous equine proteins, these antisera may cause serum sickness or anaphylaxis. A human immune serum formulation, named BabyBIG (California Department of Public Health) recently became available, but in limited supplies and to treat only infant botulism [18,28]. Additionally, the production of immune sera involves complicated and time-consuming manufacturing processes and quality management. Therefore, the development of safe, effective, and higher productive antibodies is required. Several studies have reported on development of monoclonal antibodies against various BoNT subtypes [29,30,31,32,33,34,35,36], and formulations based on these antibodies are under review by the Food Drug Administration (FDA) in US.

In this study, we generated fully human monoclonal antibodies (HuMAbs) against BoNT/B1, designated M2, M4, and S1, using a murine–human fusion partner cell line named SPYMEG [37,38,39]. We found that M2 and S1 bound to the L chain of BoNT/B1, and M4 bound to the H_C_. Furthermore, M2 showed potent neutralization activity, and the combination of M2 and M4 showed a synergistic neutralization effect. Moreover, this combination provided complete protection against BoNT/B1 in models of both botulism treatment (post-exposure) and prevention (pre-exposure). Finally, we confirmed that this combination also possessed neutralization activity against subtypes BoNT/B2 and BoNT/B6. These results indicate that the combination of M2 and M4 may serve as an effective therapeutic agent for BoNT/B intoxication.

## 2. Results

### 2.1. Immunization with Tetravalent Botulinum Toxoid Vaccine

Two healthy adult volunteers were inoculated four or five times with a tetravalent botulinum toxoid. At 9 and 18 days after the last vaccination, peripheral blood samples were collected from each volunteer. Plasma antibody titers against BoNT/A1 or BoNT/B1 were measured by enzyme-linked immunosorbent assay (ELISA). In both volunteers, the antibody titers were higher than that of non-vaccinated volunteers as previously reported [40] (Table 1).

### 2.2. Preparation of HuMAbs

Peripheral blood mononuclear cells (PBMCs) were isolated from peripheral blood samples and fused with SPYMEG cells. After HAT selection, ELISA (first screening) was performed using plates coated with a mixture of BoNT/A1 and BoNT/B1. Twenty-seven and eight positive wells, respectively, were obtained from blood samples collected 9 and 18 days after the last vaccination. These wells were subjected to cell cloning by limiting dilution. Finally, we obtained eight stable hybridoma clones. Isotype analysis showed that four hybridoma clones, designated M1, M2, M4, and S1, produced IgG; three hybridoma clones, designated M3, M5, and M6, produced IgM; and one hybridoma clone, designated M7, produced IgA. Furthermore, an IgG subclass assay revealed that M2, M4, and S1 were each comprised of an IgG1 H chain and L (lambda) chain (Table 2).

### 2.3. Binding Specificity of HuMAbs

We selected three IgG1 HuMAbs, namely M2, M4, and S1, and analyzed the binding activity of these HuMAbs against BoNT/A1 and BoNT/B1 by ELISA. All three HuMAbs bound to BoNT/B1, whereas none bound to BoNT/A1 (Figure 1). Each EC_50_ (HuMAb concentration which showed 50% of the peak signal on a serial dilution in ELISA) binding affinity to BoNT/B1 were M2: 22.5 ng/mL, M4: 15.3 ng/mL, and S1: 118 ng/mL. We next performed ELISA using recombinant proteins to determine whether the HuMAbs recognized the L and H chains (H_N_, H_C_) of BoNT/B1. M2 and S1 bound to the L chain of BoNT/B1, while M4 bound to the H_C_. All three HuMAbs did not bind to H_N_ (Figure 2A). To further investigate if the epitopes recognized by HuMAbs overlapped, we performed competitive ELISA. Non-labeled M2 and S1 were equally effective at inhibiting the binding of HRP-labeled M2 to BoNT/B1 and that of binding of HRP-labeled S1 to BoNT/B1, suggesting that M2 and S1 recognized overlapping epitopes. By contrast, binding of HRP-labeled M4 was not inhibited by non-labeled M2 and S1, indicating that M4 recognized a different epitope than M2 and S1 (Figure 2B).

### 2.4. Neutralization Activity of HuMAbs

The neutralization activity of HuMAbs against BoNT/B1 was determined by mouse bioassay. A dose of 10 mouse intraperitoneal (i.p.) lethal dose 50% (LD_50_) of BoNT/B1 was incubated with 100 µg of HuMAb for 1 h prior to i.p. injection into mice. Control mice treated with PBS died. By contrast, mice treated with M4 were partially protected, as evidenced by an increased time-to-death compared with control mice. Furthermore, complete protection was observed in mice treated with M2 (Figure 3A). Because the S1-producing hybridoma had low productivity, we could not obtain a sufficient quantity of S1 to test its neutralization activity in the mouse bioassay.

It is known that a combination of monoclonal antibodies shows a synergistic effect neutralizing BoNT [29,30,31,32,33,34,35,36,41,42,43]. Thus, we next tested for potential synergy of a combination of HuMAbs. A dose of 10 i.p. LD_50_ of BoNT/B1 was incubated with two combinations of HuMAbs (5.0 µg each) and injected into mice. The combination of M2 and S1 showed a very weak effect, and the combination of M4 and S1 showed partial protection. By contrast, all of the mice that received the combination of M2 and M4 survived and had no symptoms (Figure 3B). Furthermore, the combination of M2 and M4 (0.5 µg + 0.5 µg) showed complete protection, although the neutralization activities of single administration (0.5 µg) were partially, indicating that this combination acted highly synergistically in neutralizing BoNT/B1 (Figure 3C). Moreover, the combination of M2 and M4 (1.25 µg + 1.25 µg) showed complete protection against 80 i.p. LD_50_ (Figure 3D). All surviving mice showed no symptoms in the observation period (2 weeks). Based on these results, we used M2 and M4 in subsequent experiments. We obtained the sequences of the variable domains of the H and L chains of these antibodies using reverse transcription polymerase chain reaction (RT-PCR) with consensus primers (Figure 4).

BoNTs are always produced along with one or more neurotoxin-associated proteins that non-covalently associate with the BoNT to form progenitor toxin complexes (PTCs). *C. botulinum* type B strains produce the progenitor toxins M-PTC and L-PTC [44]. M-PTC is composed of BoNT and non-toxic non-HA (NTNHA), whereas L-PTC consists of BoNT, NTNHA, and haemagglutinin (HA). Food-borne botulism and infant botulism are caused by intestinal absorption of these toxin complexes. Therefore, analysis of the neutralization effect of HuMAbs using progenitor toxins is important for development of therapeutic agents for botulism. We tested the therapeutic effect of HuMAbs against orally administrated L-PTC in mice (a post-exposure treatment mouse model). The symptoms of botulism in mice, including fuzzy hair, muscle weakness, and respiratory failure, were observed at 12–24 h after oral ingestion of L-PTC, and untreated mice died. By contrast, complete survival was observed following M2+M4 (0.5 µg each) treatment at 12 h after oral administration of L-PTC, and partial survival was observed when treatment was given at 24 and 36 h (Figure 5A). Furthermore, we tested the preventive effect of passive immunization with HuMAbs (a pre-exposure prevention mouse model). In this model, mice were administered i.p. BoNT/B1 (10 i.p. LD_50_). All non-treated mice died. By contrast, pre-administration with M2+M4 (0.5 µg each) up to 3 days before injection of BoNT/B1 resulted in complete protection against the lethal dose of BoNT/B1. Partial protection was observed in mice administered M2+M4 at 5 or 7 days before BoNT injection (Figure 5B). Changes of botulism symptoms in each mouse in the period of observation are shown in Appendix A.

### 2.5. Binding and Neutralization Activity of HuMAbs against Subtypes BoNT/B2 and BoNT/B6

We next tested the breadth of the neutralization activity of M2+M4 by testing their efficacy against subtypes BoNT/B2 and BoNT/B6, which caused infant botulism in Japan [45,46]. ELISA showed that M2 and M4 bound strongly to BoNT/B2 and BoNT/B6 as well as BoNT/B1 (Figure 6A). In the neutralization test, mice injected with 10 ng of BoNT/B2 or 2.5 ng of BoNT/B6 died within 12 h. M2+M4 (0.5 µg each) completely neutralized BoNT/B2 and BoNT/B6 (Figure 6B). All surviving mice showed no symptoms in the observation period.

## 3. Discussion

Mouse or humanized monoclonal neutralizing antibodies against BoNT/B have been reported in many studies [29,31,43,47,48,49,50,51,52]. In addition, fully human monoclonal antibodies (HuMAbs) have been shown to effectively neutralize BoNT/B [35,36,51,52], and when used therapeutically in humans, are considered to carry a lower risk of side effects such as serum sickness and anaphylaxis than mouse, humanized, or equine antibodies. Thus, HuMAb therapy may be an effective treatment for human botulism. In this study, we developed fully human monoclonal antibodies specific for BoNT/B using the SPYMEG cell line. We developed nine hybridoma which are producing antibodies against BoNT/B1. We could not obtain BoNT/A1 specific antibody producing hybridoma in this screening. Although the cause of this result is unknown. It could be related to diatheses of volunteers or screening methods. We selected IgG1 (M2, M4, and S1) because IgG1 is the most commonly used subtype for antibody formulations. Among HuMAbs against BoNT/B1, M2 specifically bound to the L chain of BoNT/B1 and showed potent neutralization activity in a mouse bioassay (approximately 10 i.p. LD_50_/100 µg of antibody) (Figure 3A). Mechanistically, M2 may directly inhibit the proteolytic activity of the L chain or prevent the translocation of the L chain into the cytosol. In fact, both of these mechanisms were reported in a HuMAb that recognized the L chain of BoNT/A [52]. On the other hand, M4 bound to the H_C_ (receptor binding domain) and showed partial neutralization activity. M4 may prevent the binding of BoNT/B1 to neuronal cells [29]. Meanwhile, the reactivity of M4 with BoNT/B1 was very weak in western blot analysis (denaturing condition) (data not shown). Hence, M4 may recognize the conformational epitope of BoNT/B1. The epitope binding details and neutralization mechanisms of M2 and M4 are currently being analyzed. We are planning to reveal the step of the botulinum neurotoxin mechanism of action that is inhibited by M2 and M4 using neuronal cells, such as PC12 cells.

Several studies have confirmed a synergistic effect when two or more antibodies are combined [29,30,31,32,33,34,35,36,41,42,43]. In the case of BoNT/B neutralization, to take an example, combination of three mouse monoclonal antibodies (total 2.5 µg) showed complete protection against 80 LD_50_ [48]. We analyzed the neutralization activity of two combinations of HuMAbs to determine the combination with the most potent neutralization activity (Figure 3B). The combination of the HuMAbs (M2+M4), which recognize non-overlapping epitopes, had the most potent neutralization activity among the combinations of M2, M4, and S1, and was able to completely neutralize BoNT/B1 with a potency of 10 i.p. LD_50_/1.0 µg of antibodies (Figure 3C) and 80 i.p. LD_50_/2.5 µg of antibodies (Figure 3D). From these results, the combination of “two” HuMAbs, M2+M4 showed neutralization activity against BoNT/B1 approximately equal to the antibodies previously reported which consist of three monoclonal antibodies [48]. It has been suggested that a combination of multiple (three or more) HuMAbs is required to effectively neutralize BoNT/B [35,36,53]. In contrast, two HuMAbs combination, M2+M4 exhibited sufficiently high neutralization activity against BoNT/B1 by the synergistic effect at least in our mouse models. The mechanism of the synergistic effect is unknown. However, the combination of L chain-binding antibody and H_C_-binding antibody might be important for the synergistic effect. By contrast, the combination of M2 and S1, which recognize overlapping epitopes, showed no synergistic effect. This result suggests that the recognition of non-overlapping epitopes is critical for a synergistic effect in BoNT neutralization. Meanwhile, the combination of M4 and S1 showed only partial protection even though M4 and S1 recognize non-overlapping epitopes. The antigen binding affinity of S1 was lower than that of M2 (Figure 1), or the amino acid sequence of BoNT/B1 recognized by S1 might be slightly different from that of M2, both of which are factors that may have influenced the synergistic effect of M4+S1.

Food-borne botulism and infant botulism, which account for the majority of cases of human botulism, are caused by the intestinal absorption of the progenitor toxins M-PTC and L-PTC. The oral toxicity of L-PTC is much higher than the toxicity of M-PTC and BoNT individually [19,54]. Thus, L-PTC is considered to be the predominant contributor to the onset of food-borne botulism and presumably infant botulism. In human food-borne botulism, the signs and symptoms of botulism typically begin between 12 and 36 h after ingestion of the toxin [18]. Therefore, we used a dose of L-PTC at which almost all control mice presented with botulism symptoms approximately 12–24 h after oral administration of L-PTC. In this mouse model, M2+M4 provided complete protection when administered within 12 h of oral ingestion of L-PTC, and partial survival was observed at 24 and 36 h after L-PTC ingestion. Our results show that M2+M4 exerts potent protective activity even after botulism symptoms have developed, but early administration is important. In fact, in the case of food-borne botulism, it is recommended to administer antitoxin as early as possible after toxin exposure. Good treatment outcomes are directly correlated with early administration of antitoxins [18]. Preventive countermeasures against BoNT are also needed for individuals at risk of BoNT exposure, such as first responders in contaminated areas (for example, in instances of bioterrorism or outbreak). Previous studies have reported that passive immunization with HuMAbs protects against the onset of type A botulism [55,56]. In this study, M2+M4 showed a long-term preventive effect against a lethal dose of BoNT/B1 (Figure 5B). It is presumed that because of their low immunogenicity and high affinity to human neonatal Fc receptor compared with mouse antibodies, the HuMAbs M2 and M4 may exist longer in the human bloodstream and therefore neutralize BoNT/B1 more effectively and for a longer period of time [57]. Taken together, these data indicate that M2+M4 has sufficient therapeutic and preventive effects against botulism to permit practical use.

We also examined the neutralization effect of M2+M4 against other BoNT/B subtypes. For BoNT/B, currently eight subtypes, B1 to B8 are known, with most BoNT/B strain producing B1 and B2 [58,59,60]. Therefore, we selected BoNT/B2 as an analysis target in addition to BoNT/B1. Furthermore, we also selected BoNT/B6 which caused human botulism similarly to BoNT/B1 and BoNT/B2. The BoNT/B1-producing strain Okra was isolated from a case of food-borne botulism [61]. The BoNT/B2-producing strain 111 and the BoNT/B6-producing strain Osaka05 were isolated from infant botulism cases in Japan [45,46]. BoNT/B2 and BoNT/B6 exhibit toxicies in mice is one-tenth and one-fifth as toxic as BoNT/B1 (BoNT/B2: 1.71 × 10^7^ i.p. LD_50_/mg protein, BoNT/B6: 2.55 × 10^7^ i.p. LD_50_/mg protein) [62]. BoNT/B2 and BoNT/B6 show different antigenic and biological properties than BoNT/B1 [45,46,61,63]. However, M2 and M4 exhibited strong binding and high neutralization activity against BoNT/B2 and BoNT/B6 as well as against BoNT/B1 (Figure 6). From these results and previous report [62], neutralization activity of M2+M4 is considered to be approximately 171 i.p. LD_50_/1.0 µg of antibodies against BoNT/B2, and approximately 63.75 i.p. LD_50_/1.0 µg of antibodies against BoNT/B6. This neutralization activity of M2+M4 against BoNT/B2 may be greater than the antibodies previously reported (BoNT/B2 complexed form, 5.0 LD_50_/2.5 µg + 2.5 µg two combination of IgG) [50]. These data suggest that M2 and M4 may be effective for the treatment and prevention of botulism caused by broad subtypes of BoNT/B. Meanwhile, BoNT/B4, produced by non-proteolytic *C. botulinum* (group II), is the most dissimilar among all the BoNT/B subtypes so far and differs from BoNT/B1 by 6.8% at the amino acid level [60]. Further analysis on the neutralization effect against other subtypes of BoNT/B including BoNT/B4 would strengthen this proposition.

## 4. Conclusions

We developed anti-BoNT/B antibody–producing hybridomas using the SPYMEG cell line, and obtained two neutralizing antibodies, M2 and M4. This method can be applied to other botulinum serotypes, and it will be possible to maintain a stable supply of HuMAbs using these hybridomas. Because of their human origin, M2 and M4 are safe for the treatment of human botulism, including infant botulism. Furthermore, we found that the combination of M2 and M4 had potent neutralization activity against BoNT/B1 in spite of only two HuMAbs, and neutralized two other subtypes. Additionally, M2+M4 showed therapeutic and preventive effects against botulism in mouse models. These data indicate that the fully human monoclonal antibodies, M2 and M4, are one of the promising candidates for the development of human therapeutics and prophylactics for BoNT/B intoxication.

## 5. Materials and Methods

### 5.1. Ethics Statement

Human materials were collected using protocols approved by the Institutional Review Board of the Osaka University Research Institute for Microbial Diseases (#20-8, approved on 8 April 2009). Written informed consent was obtained from the participants. Animal studies were conducted under the applicable laws and guidelines for the care and use of laboratory animals at the Osaka University Research Institute for Microbial Diseases and Kanazawa University. They were approved by the Animal Experiment Committee of the Osaka University Research Institute for Microbial Diseases (#H21-27-0, approved on 19 February 2010, #H27-02-0, approved on 30 April 2015) and Kanazawa University (AP-163710, approved on 15 March 2016).

### 5.2. Preparation of BoNT/A and BoNT/B

*C*. *botulinum* serotype B1 strain Okra was cultured for 5 days at 35 °C using a cellophane tube procedure [dialysis outer liquid: 3% pepton (Sigma-Aldrich, St. Louis, MO, USA), 6% polypepton (FUJIFILM, Osaka, Japan), 6% lactalbumin hydrolysate (FUJIFILM), 3% yeast extract (Becton, Dickinson and Company, Sparks, MD, USA), dialysis inner liquid: 10% glucose (FUJIFILM), 5% NaCl (FUJIFILM), 1% L-cysteine hydrochloride monohydrate (FUJIFILM)], and the culture supernatant was obtained. The culture supernatant was concentrated by 60% ammonium sulfate precipitation. After dialysis, progenitor toxins (M-PTC and L-PTC) were purified from the culture supernatant using an SP-Toyopearl 650 M column (Tosoh, Tokyo, Japan) and a lactose gel column (EY Laboratories, San Mateo, CA, USA). BoNT/B1 and a non-toxic component were prepared from L-PTC using the lactose gel column under alkaline conditions of pH 8.0, as described previously [64]. The lethal toxicity of BoNT/B1 was determined using mouse bioassay [65,66]. BoNT/A1 (strain 62A), BoNT/B2 (strain 111), and BoNT/B6 (strain Osaka05) were provided by Dr. T. Kohda (Osaka Prefecture University).

### 5.3. Inoculation of Botulinum Toxoid Vaccine

Two healthy adult volunteers were inoculated four or five times with a tetravalent botulinum toxoid vaccine (types A, B, E, and F) [40] provided by Dr. M. Takahashi (National Institute of Infectious Diseases). The toxoid was injected intramuscularly in 0.5-mL doses. Blood samples were collected from each volunteer and the plasma antibody titers against BoNT/A1 and BoNT/B1 were tested by ELISA.

### 5.4. Preparation of HuMAbs

Peripheral blood samples (10 mL) were collected from volunteers 9 or 18 days after the last immunization, and PBMCs were purified by Ficoll (GE Healthcare, Buckinghamshire, UK) gradient centrifugation. The PBMCs were fused with cells from the mouse–human fusion partner cell line SPYMEG (Medical & Biological Laboratories, Nagoya, Japan) at a ratio of 10:1 with polyethylene glycol (Roche, Basel, Switzerland). Fused cells were cultured in Dulbecco’s modified Eagle medium (DMEM, Thermo Fisher Scientific, Waltham, MA, USA) supplemented with 15% heat-inactivated fetal bovine serum in 96-well plates for 10–14 days in the presence of hypoxanthine–aminopterin–thymidine (HAT, Thermo Fisher Scientific). The first screening of the culture medium for antibodies specific to BoNT/A1 or BoNT/B1 was performed by ELISA. BoNT/A1 or BoNT/B1-specific antibody-positive wells were next subjected to cell cloning by limiting dilution. Hybridomas were collected, diluted in medium, and then used to seed new plates (1–2 cells/well) to obtain a monoclonal cell population. The plates were incubated for 10–14 days. The second screening was also performed by ELISA. After cell cloning, cultures of BoNT/A1 or BoNT/B1-specific antibody producing-hybridomas were scaled up and we made cell stocks. Each stable hybridoma was cultured in serum-free medium (Thermo Fisher Scientific), and then the supernatant was collected. Isotypes of HuMAbs in culture supernatant were determined by western blot using anti-human IgG conjugated with HRP (Bio-Rad Laboratories, Berkeley, CA, USA), anti-human IgM conjugated with HRP (Invitrogen, Carlsbad, CA, USA), or anti-human IgA conjugated with HRP (Invitrogen). IgG antibodies were purified from the supernatant using a Protein G column (GE Healthcare). The concentration of IgG was determined by BCA Protein Assay Kit (Thermo Fisher Scientific). The subclass of each IgG antibody was determined using a human IgG subclass ELISA kit (Invitrogen).

### 5.5. Preparation of Recombinant Proteins

Purified DNA from *C. botulinum* type B strain Okra was used as a template for the amplification of DNA encoding L chain (aa 1-430), H_N_ (aa 449-858), and H_C_ (aa 858-1291) by PCR. The amplified DNAs were inserted into *Nco*I- *Xho*I site of the pET28b vector (Novagen, Merck Millipore, Madison, WI, USA). Recombinant proteins were expressed as C-terminally His-tagged proteins in *E. coli* strain BL21-CodonPlus (DE3)-RIL (Agilent Technologies, Santa Clara, CA, USA). Protein expression was induced using Overnight Express^TM^ Autoinduction System 1 (Merck Millipore). *E. coli* were cultured for 48 h at 18 °C or 36 h at 30 °C, and then were harvested. Bacterial cells were sonicated, and purified recombinant proteins from extracts using HisTrap HP (GE Healthcare). The concentration of recombinant protein was determined by BCA Protein Assay Kit.

### 5.6. Binding Assay Using ELISA

The 96-well plates (Corning, Corning, NY, USA) were coated with BoNT, recombinant L chain, recombinant H_N_, or recombinant H_C_ (300 ng/well) for 2 h at 37 °C. The wells were washed three times with phosphate-buffered saline (PBS) containing 0.05% Tween20 (Sigma-Aldrich) (PBS-T) and blocked with 0.2% bovine serum albumin (BSA, Sigma-Aldrich)/PBS-T overnight at 4 °C. Human plasma samples or HuMAbs were added to a well and incubated for 2 h at 37 °C. After washing three times, anti-human IgG conjugated with HRP (Bio-Rad Laboratories or Jackson ImmunoResearch, West Grove, PA, USA) was added and incubated for 2 h at 37 °C. Plates were washed three times again and then incubated with substrate solution (*o*-phenylendiamine, Nacalai Tesque, Kyoto, Japan or ABTS, Roche) for 20 min at 37 °C, and absorbance values at Abs_492_ or Abs_405_ were measured, respectively.

### 5.7. Competition ELISA

M2, M4, and S1 were labeled with HRP using Perioxidase Labeling Kits (Dojindo Molecular Technologies, Kumamoto, Japan). The optimal dilutions of HRP-labeled HuMAbs were determined by ELISA based on an Abs_405_ value around 2.0. Non-labeled HuMAb (2.5 µg/mL) was added to plates coated with BoNT/B1 and incubated for 2 h at 37 °C. After washing, HRP-labeled M2 (1:5000), M4 (1:5000), or S1 (1:1000) was added and incubated for 2 h at 37 °C. Plates were washed again and incubated with substrate solution (ABTS) for 20 min at 37 °C, and absorbance values at Abs_405_ were measured.

### 5.8. Sequencing of HuMAb Variable Region Gene Segments

Total RNA was extracted from the hybridomas using an RNeasy Mini Kit (Qiagen, Hilden, Germany), and cDNA was synthesized by RT-PCR using a SuperScript^®^ VILO^TM^ cDNA Synthesis Kit (Invitrogen). The coding regions of the H and L chains of M2 and M4 were amplified by PCR using KOD-Plus-Neo (TOYOBO, Osaka, Japan), with the following primers: 5′-ATGGACTGGACCTGGAGGATCCTC-3′ (M2 H chain sense primer), 5′-ATGAAACACCTGTGGTTCTTCCTCCT-3′ (M4 H chain sense primer), and 5′-CTCCCGCGGCTTTGTCTTGGCATTA-3′ (H chain antisense primer); and 5′-ATGSCCTGGGCTCYKCTSCTCCTS-3′ (M2 L chain sense primer), 5′-ATGGCCTGGWYYCCTCTCYTYCTS-3′ (M4 L chain sense primer), and 5′-TGGCAGCTGTAGCTTCTGTGGGACT-3′ (L chain antisense primer).

### 5.9. Neutralization Test (Mouse Bioassay)

The neutralization activity of HuMAbs was tested by mouse bioassay. Female ddY mice were purchased from SLC (Shizuoka, Japan) and were used at 4 weeks of age. 10 mM sodium phosphate buffer (pH 6.0) containing 0.1% gelatin was used for sample dilution. One or more HuMAbs was incubated with BoNT/B1, BoNT/B2, or BoNT/B6 at room temperature for 1 h prior to i.p. injection (in a total volume of 500 µL) into mice. In control mice, BoNT was incubated with PBS instead of HuMAbs. In the botulism treatment (post-exposure) model, the B1 subtype progenitor toxin (L-PTC, 10 ng in a volume of 300 µL) was orally administered and M2+M4 was subsequently administered at 12, 24, and 36 h after oral administration of L-PTC by i.p. injection. In the botulism prevention (pre-exposure) model, M2+M4 was administered by i.p. injection and mice were then challenged at 1, 3, 5, or 7 days later with 10 i.p. LD_50_ BoNT/B1 by i.p. injection. Mice were observed for morbidity and mortality for 2 weeks.

## Figures and Tables

**Figure 1 toxins-12-00302-f001:**
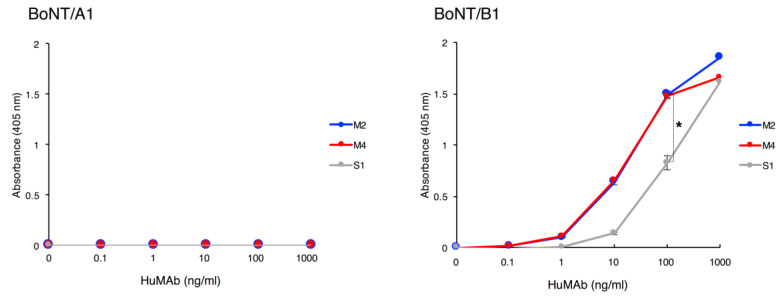
Binding of HuMAbs to BoNT/A1 and BoNT/B1. Binding of HuMAbs to BoNT was analyzed by ELISA. HuMAbs (0.1–1000 ng/mL) were added to plates coated with BoNT/A1 or BoNT/B1. After washing, bound HuMAbs were detected by anti-human IgG antibody conjugated with HRP. Error bars indicate s.d. Statistical analyses were performed with one-way ANOVA followed by Tukey’s multiple comparison test (* *p*-value < 0.01). Data are representative of three independent experiments.

**Figure 2 toxins-12-00302-f002:**
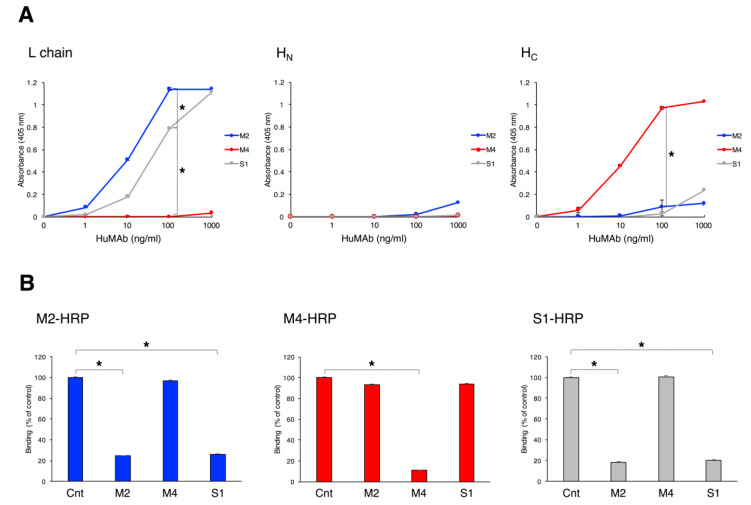
Binding domains of BoNT/B1 recognized by HuMAbs. (**A**) Binding of HuMAbs to recombinant L chain, H_N_, and H_C_ were analyzed by ELISA. HuMAbs (1.0–1000 ng/mL) were added to plates coated with recombinant L chain, H_N_, or H_C_. After washing, bound HuMAbs were detected by anti-human IgG antibody conjugated with HRP. Error bars indicate s.d. Statistical analyses were performed with one-way ANOVA followed by Tukey’s multiple comparison test (* *p*-value < 0.01). Data are representative of three independent experiments. (**B**) Binding epitopes of HuMAbs were confirmed by competitive binding ELISA. Non-labeled HuMAbs (2.5 µg/mL) were added to plates coated with BoNT/B1, and then HRP-labeled M2 (1:5000), M4 (1:5000), or S1 (1:1000) was added. Bound HRP–labeled HuMAbs were detected. Results are expressed as percent binding of each HRP-labeled HuMAb, with the 100% value defined by binding in the absence of non-labeled HuMAbs. Error bars indicate s.d. Statistical analyses were performed with the Student’s *t*-test (* *p*-value < 0.01). Data are representative of two independent experiments.

**Figure 3 toxins-12-00302-f003:**
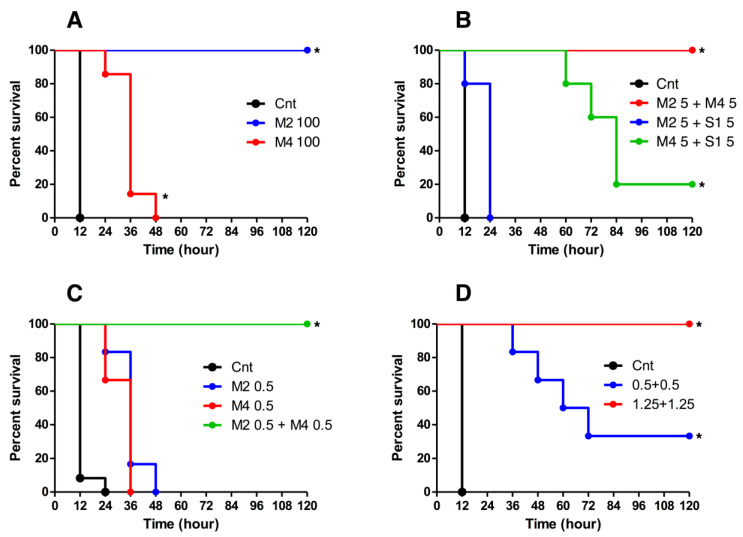
Neutralization activity of HuMAbs against BoNT/B1. The neutralization activity of HuMAbs against BoNT/B1 was determined by mouse bioassay. (**A**) A dose of 10 i.p. LD_50_ of BoNT/B1 was incubated with 100 µg of M2 or M4 for 1 h prior to i.p. injection into mice. Control mice (Cnt) were treated with PBS instead of HuMAbs. Mice were observed for morbidity and mortality for 2 weeks. *n* = 5 per group. (**B**) Neutralization of BoNT/B1 with two combinations of HuMAbs. A dose of 10 i.p. LD_50_ of BoNT/B1 was incubated with a mixture of HuMAbs (5.0 µg each) and administered by i.p. injection into mice. Control mice (Cnt) were treated with PBS instead of HuMAbs. Mice were observed for morbidity and mortality for 2 weeks. Cnt, *n* = 10 per group, M2+M4, *n* = 5 per group, M2+S1, *n* = 5 per group, M4+S1, *n* = 5 per group. (**C**) Synergistic effect of M2+M4. A dose of 10 i.p. LD_50_ of BoNT/B1 was incubated with a M2 (0.5 µg), M4 (0.5 µg), or mixture of M2 and M4 (0.5 µg each) and then administered by i.p. injection into mice. Control mice (Cnt) were treated with PBS instead of HuMAbs. Mice were observed for morbidity and mortality for 2 weeks. Cnt, *n* = 12 per group, M2 0.5 µg, *n* = 6 per group, M4 0.5 µg, *n* = 6 per group, 0.5 µg each, *n* = 7 per group. (**D**) Neutralization of BoNT/B1 with M2+M4. A dose of 80 i.p. LD_50_ of BoNT/B1 was incubated with a mixture of HuMAbs (0.5 µg each or 1.25 µg each) and administered by i.p. injection into mice. Control mice (Cnt) were treated with PBS instead of HuMAbs. Mice were observed for morbidity and mortality for 2 weeks. Cnt, *n* = 6 per group, M2+M4 (0.5 µg each), *n* = 6 per group, M2+M4 (1.25 µg each), *n* = 6 per group. The Kaplan–Meier estimator method was used to draw survival curves, and a log-rank test was used to compare the survival rates (* *p*-value < 0.001).

**Figure 4 toxins-12-00302-f004:**
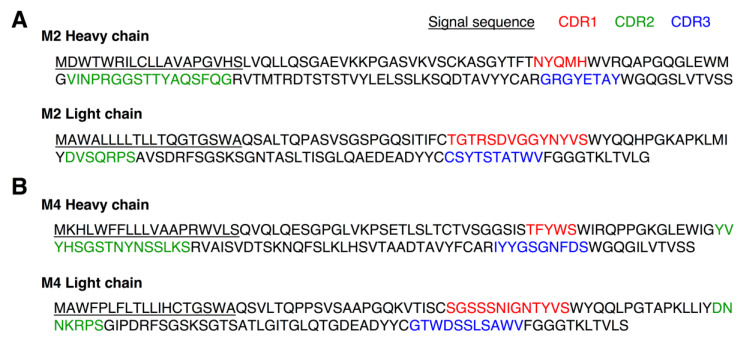
Amino acid sequences of the variable regions of the heavy and light (lambda) chains of (**A**) M2 and (**B**) M4. Complementarity-determining regions (CDRs) are indicated in red (CDR1), green (CDR2), and blue (CDR3). Signal sequences are underlined.

**Figure 5 toxins-12-00302-f005:**
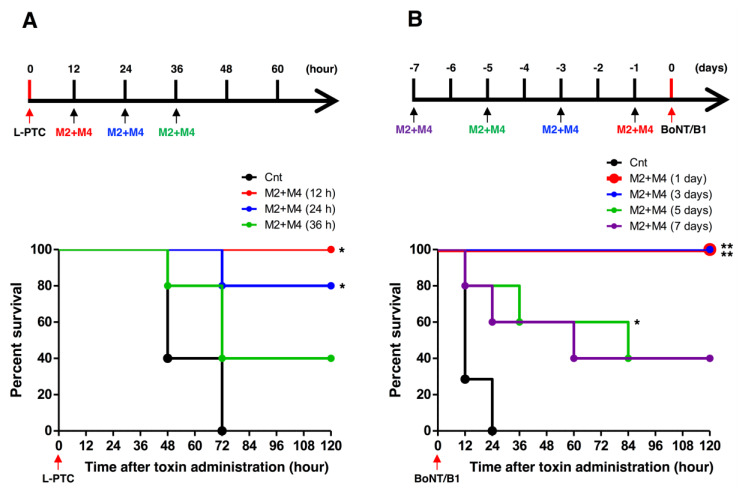
Therapeutic and preventive effects of HuMAbs (M2+M4) against BoNT/B1 intoxication. (**A**) Mice were orally administered progenitor toxin (L-PTC, 10 ng), and subsequently administered M2+M4 (0.5 µg each) by i.p. injection at 12, 24, or 36 h after the oral administration of L-PTC. Control mice (Cnt) were not treated with HuMAbs. Mice were observed for morbidity and mortality for 2 weeks. *n* = 5 per group. (**B**) Mice received i.p. injection of M2+M4 (0.5 µg each) and were then challenged at 1, 3, 5, or 7 days later with i.p. administration of 10 i.p. LD_50_ BoNT/B1. Control mice (Cnt) were not injected with HuMAbs. Mice were observed for morbidity and mortality for 2 weeks. Cnt, *n* = 7 per group, M2+M4 (1 day), *n* = 5 per group, M2+M4 (3 days), *n* = 5 per group, M2+M4 (5 days), *n* = 5 per group. M2+M4 (7 days), *n* = 5 per group. The Kaplan–Meier estimator method was used to draw survival curves, and a log-rank test was used to compare the survival rates (* *p*-value < 0.01, ** *p*-value < 0.001).

**Figure 6 toxins-12-00302-f006:**
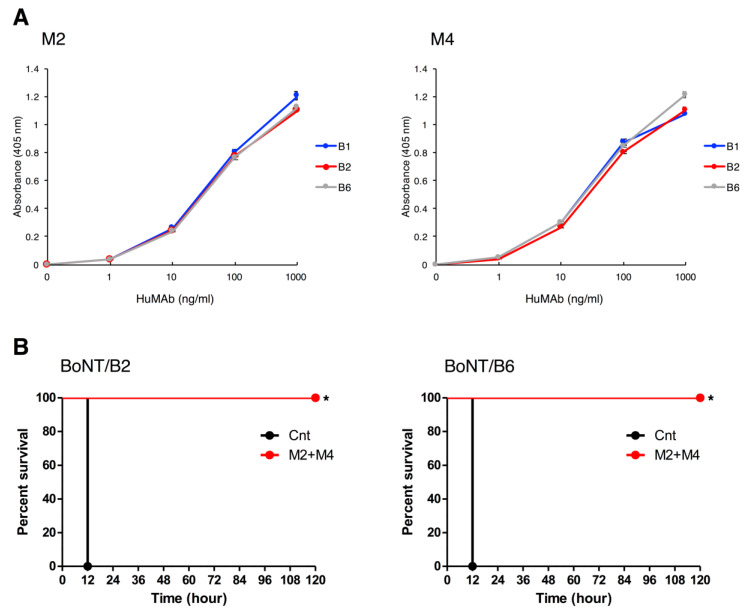
Binding and neutralization activity of HuMAbs against BoNT/B2 and BoNT/B6. (**A**) Binding of HuMAbs to BoNT/B2 and BoNT/B6 were analyzed by ELISA. HuMAbs (1.0–1000 ng/mL) were added to plates coated with BoNT/B1, BoNT/B2, or BoNT/B6. After washing, bound HuMAbs were detected by anti-human IgG antibody conjugated with HRP. (**B**) Doses of 10 ng of BoNT/B2 or 2.5 ng of BoNT/B6 were incubated with M2+M4 (0.5 µg each) and administered by i.p. injection into mice. Control mice (Cnt) were treated with PBS instead of HuMAbs. Mice were observed for morbidity and mortality for 2 weeks. *n* = 5 per group. The Kaplan–Meier estimator method was used to draw survival curves, and a log-rank test was used to compare the survival rates (* *p*-value < 0.001).

**Table 1 toxins-12-00302-t001:** Antibody titers against BoNT/A1 or BoNT/B1 in plasma samples from volunteers. Antibody titers against BoNT/A1 or BoNT/B1 from the plasma samples of human volunteers were tested using ELISA. Plasma samples were serially diluted two-fold and 50 µL of each dilution was added to plates coated with BoNT/A1 or BoNT/B1. After washing, bound HuMAbs were detected by anti-human IgG antibody conjugated with horseradish peroxidase (HRP). ELISA titers are expressed as the highest dilution factor with an absorbance at least twice that of negative control plasma obtained from non-vaccinated volunteers.

Volunteer Number	Number of Vaccinations	Days after Last Immunization	ELISA Titer (log_2_) BoNT/A1	ELISA Titer (log_2_) BoNT/B1
Bkf002	5	9	14	13
Bkf002	5	18	13	15
Bkf003	4	9	13	12
Bkf003	4	18	13	14

**Table 2 toxins-12-00302-t002:** Isotypes of HuMAbs. ^*^ The IgG subclass of M1 could not be determined with the IgG subclass human ELISA kit.

Volunteer Number	Days after Last Immunization	Clone No.	Isotype
Bkf002	9	M1	IgG ^*^
	9	M2	IgG1
	9	M3	IgM
	9	M4	IgG1
	9	M5	IgM
	9	M6	IgM
	18	M7	IgA
Bkf003	9	S1	IgG1

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
