# Peer review of "Fully Human Monoclonal Antibodies Effectively Neutralizing Botulinum Neurotoxin Serotype B"

_toxins, 2020, doi:10.3390/toxins12050302_

Round 1

Reviewer 1 Report

The authors generated human monoclonal antibodies (HuMAbs) against BoNT/B. They showed that 2 of them specifically bind the LC, M2 or the HC, M4 of BoNT/B. The in vivo data presented nicely showed that these antibodies are effective in inhibit botulism in vivo using several paradigms (both prophylactically and therapeutically). The authors also demonstrated that these antibodies are effective in neutralizing BoNT/B1, /B2 and B6. In summary, the paper is very interesting however prior publication several points should be adressed:

1) More recent reviews and articles on serotypes classification and botulinum neurotoxins mechanism of action should be citied and discussed. Please take in consideration that (i) more than seven serotypes of BoNTs have been discovered and (ii) the thioredoxin/thioredoxin reductase redox system together with the Hsp90 chaperon are of pivotal importance for LC translocation into the cytosol.  Please refer to: (i) Zornetta I. et al., Scientific Reports, 2016; (ii) Azarnia Tehran D. et al., Toxins 2018; (iii) Pirazzini M et al., Toxicon 2018; (iv) Zhang et al., Nature Comm,  2017; (v) Anniballi F. et al., Infection Genetics and Evolution, 2016.

1) Please clarify why only three IgG1 HuMAbs (M2, M4 and S1) were selected for further analysis.

2) In Fig. 1 and in Fig.2A standard deviation (derived from 3 independent experiments, as reported by authors) and statistical analysis should be reported.

4) In all experiments were the use of HuMAbs results in total protection (100% of survival), the authors should explain if mice showed any symptoms of botulism even after the time window reported. Moreover, in all graphs statistical analysis is missing (see all figures along the paper).

5) Fig. 3: in all graphs the concentrations of HuMAbs used should be indicated (as in Fig.3C and 3D).

5) Fig. 3C: In order to verify if the three different HuMAbs have a synergistic effect, data showing the single protective (or not) effect of S1 using (5.0 μg) should be reported.

6) Fig.3B and Fig.3C are difficult to understand for the readers. The synergistic effect of M2+M4 in Fig.3B is the same of Fig.3C (it is the same experiment)? Please think to combine the two graphs.

7) Fig. 5: the authors stated that mice were observed for morbidity and mortality for 2 weeks. Please explain if mice showed any symptoms of botulism after 120 hours.

8) Fig.5: the authors should modify the graphs. It is clear from the figure legend but it is not clear in the graph that the hours (5A) or days indicated (5B) refer to the BoNT/B injection or HuMAbs administration.

9) In vitro data, using neurons, showing which step of botulinum neurotoxin mechanism of action is inhibited by the different antibodies will increase the novelty of the entire paper. 

Author Response

Dear Reviewer 1,

Re.: “Fully human monoclonal antibodies effectively neutralizing botulinum neurotoxin serotype B” by Takuhiro Matsumura, Sho Amatsu, Ryo Misaki, Masahiro Yutani, Anariwa Du, Tomoko Kohda, Kazuhito Fujiyama, Kazuyoshi Ikuta, and Yukako Fujinaga .

We are grateful for the helpful comments from you on the original version of our manuscript. We have addressed all of the comments as described below, and we hope that the explanations and revisions of our work are satisfactory.

The authors generated human monoclonal antibodies (HuMAbs) against BoNT/B. They showed that 2 of them specifically bind the LC, M2 or the HC, M4 of BoNT/B. The in vivo data presented nicely showed that these antibodies are effective in inhibit botulism in vivo using several paradigms (both prophylactically and therapeutically). The authors also demonstrated that these antibodies are effective in neutralizing BoNT/B1, /B2 and B6. In summary, the paper is very interesting however prior publication several points should be adressed:

1) More recent reviews and articles on serotypes classification and botulinum neurotoxins mechanism of action should be citied and discussed. Please take in consideration that (i) more than seven serotypes of BoNTs have been discovered and (ii) the thioredoxin/thioredoxin reductase redox system together with the Hsp90 chaperon are of pivotal importance for LC translocation into the cytosol.  Please refer to: (i) Zornetta I. et al., Scientific Reports, 2016; (ii) Azarnia Tehran D. et al., Toxins 2018; (iii) Pirazzini M et al., Toxicon 2018; (iv) Zhang et al., Nature Comm, 2017; (v) Anniballi F. et al., Infection Genetics and Evolution, 2016.

Response:

We added references that you cited and revised the sentence (lines 30-35) as follows:

Originally, seven serotypes, designated A to G, have been identified, and four of these, namely A, B, E, and F, cause human botulism [2]. Additionally, BoNT/DC, which is considered a mosaic toxin between BoNT/D and BoNT/C, has been reported. It was recently reported that novel serotype, Recently, BoNT/H was reported, and, is produced by a C. botulinum strain IBCA10-7060, which also produces BoNT/B [4,5]. subsequent studies have described that this toxin is a hybrid-toxin of BoNT/A1 and BoNT/F5 [6-8] and that its light chain and N-terminal of its heavy chain is are immunologically unique [9,10]. More recently, the novel serotype BoNT/X and BoNT-like toxin, BoNT/Wo and BoNT/En (BoNT/J), were also reported.

Additional references

  1. NakamuraK, Kohda T, Shibata Y, Tsukamoto K, Arimitsu H, Hayashi M, Mukamoto M, Sasakawa N,Kozaki S. Unique biological activity of botulinum D/C mosaic neurotoxin in murine species. Infect Immun. 2012 80, 2886-93.

  1. ZhangS, Berntsson RP, Tepp WH, Tao L, Johnson EA, Stenmark P, Dong M. Structural basis for the unique ganglioside and cell membrane recognition mechanism ofbotulinum neurotoxin DC. Nat Commun 2017, 8, 1637. 

  1. Zhang S, Masuyer G, Zhang J, Shen Y, Lundin D, Henriksson L, Miyashita SI, Martínez-Carranza M, Dong M, Stenmark P. Identification and characterization of a novel botulinum neurotoxin. Nat Commun 2017, 8, 14130.

  1. ZornettaI, Azarnia Tehran D, Arrigoni G, Anniballi F, Bano L, Leka O, Zanotti G, Binz T, Montecucco C. The first non Clostridialbotulinum-like toxin cleaves VAMP within the juxtamembrane domain. Sci Rep 2016, 6, 30257.

  1. Brunt J, Carter AT, Stringer SC, Peck MW. Identification of a novel botulinum neurotoxin gene cluster inEnterococcus. FEBS Lett 2018 592, 310-317.

  1. Zhang S, Lebreton F, Mansfield MJ, Miyashita SI, Zhang J, Schwartzman JA, Tao L, Masuyer G, Martínez-Carranza M, Stenmark P, Gilmore MS, Doxey AC, Dong M. Identification of a Botulinum Neurotoxin-like Toxin in a Commensal Strain ofEnterococcus faecium. Cell Host Microbe 2018 23, 169-176.

  1. TehranDA, Pirazzini M. NovelBotulinum Neurotoxins: Exploring Underneath the Iceberg Tip. Toxins (Basel) 2018, 10, 10.

  1. PirazziniM, Azarnia Tehran D, Zanetti G, Rossetto O, Montecucco C. Hsp90 and Thioredoxin-Thioredoxin Reductase enable the catalytic activity of Clostridial neurotoxins inside nerve terminals. Toxicon 2018, 147, 32-37.

2) Please clarify why only three IgG1 HuMAbs (M2, M4 and S1) were selected for further analysis.

Response:

IgG1 is the most commonly used subtype for antibody formulations. Therefore, we selected IgG1 (M2, M4, and S1) with the aim of practical use. We excluded M1 because the IgG subclass of M1 could not be determined. We added: “We selected IgG1 (M2, M4, and S1) because IgG1 is the most commonly used subtype for antibody formulations” (line 301).

3) In Fig. 1 and in Fig.2A standard deviation (derived from 3 independent experiments, as reported by authors) and statistical analysis should be reported.

Response:

According to the reviewer’s suggestion, we revised the figures and added: “Error bars indicate s.d.” and “Statistical analyses were performed with one-way ANOVA followed by Tukey’s multiple comparison test (*p-value <0.01)” to Fig. 1 and 2 legends.

4) In all experiments were the use of HuMAbs results in total protection (100% of survival), the authors should explain if mice showed any symptoms of botulism even after the time window reported. Moreover, in all graphs statistical analysis is missing (see all figures along the paper).

Response:

Surviving mice in Figs. 3 and 6 had no symptoms even after the time window. We added: “All surviving mice showed no symptoms in the observation period (2 weeks)” (lines 215 and 269), and performed statistical analysis in Figs. 3, 5, and 6. We revised the figures and added: “The Kaplan–Meier estimator method was used to draw survival curves, and a log-rank test was used to compare the survival rates (*p-value <0.01, **p-value <0.001)” in Fig. 3, 5, and 6 legends.

5) Fig. 3: in all graphs the concentrations of HuMAbs used should be indicated (as in Fig.3C and 3D).

Response:

This has been revised. Thank you.

6) Fig. 3C: In order to verify if the three different HuMAbs have a synergistic effect, data showing the single protective (or not) effect of S1 using (5.0 μg) should be reported.

Response:

Because the S1-producing hybridoma had low productivity, we could not obtain a sufficient quantity of S1 to test again, as stated in line 177. However, the combination of M2 and M4 exhibits the most potent neutralization activity among the combinations of M2, M4, and S1, as shown in Fig. 3B. Therefore, we focused on M2 and M4 in Figs 4, 5, and 6. We added: “had the most potent neutralization activity among the combinations of M2, M4, and S1, and” (line 329) as follows:

The combination of the HuMAbs (M2 + M4), which recognize non-overlapping epitopes, had the most potent neutralization activity among the combinations of M2, M4, and S1, and was able to completely neutralize BoNT/B1 with a potency of 10 i.p. LD50/1.0 µg of antibodies (Figure. 3C) and 80 i.p. LD50/2.5 µg of antibodies (Figure. 3D).

7) Fig.3B and Fig.3C are difficult to understand for the readers. The synergistic effect of M2+M4 in Fig.3B is the same of Fig.3C (it is the same experiment)? Please think to combine the two graphs.

Response:

We performed these mouse bioassays for a different purpose. First, we analyzed the neutralization activities of two combinations of HuMAbs to determine the combination with the most potent neutralization activity in Fig. 3B, and then demonstrated the synergistic effect of the combination of M2 and M4 in Fig. 3C. Therefore, we separated these figures. We added: “We analyzed the neutralization activity of two combinations of HuMAbs to determine the combination with the most potent neutralization activity (Figure 3B)” (line 326).

8) Fig. 5: the authors stated that mice were observed for morbidity and mortality for 2 weeks. Please explain if mice showed any symptoms of botulism after 120 hours.

Response:

Thank you for your helpful comments. All surviving mice recovered and had no symptoms after 120 h, and survived without any symptoms until 2 weeks. We generated a supplementary figure 1 showing the changes of botulism symptoms in each mouse in the period of observation (2 weeks). This figure is correlated with Figure 5. We added: “Changes of botulism symptoms in each mouse in the period of observation are shown in Figure S1” (line 247) and “Supplementary Materials: Figure S1: Therapeutic and preventive effects of HuMAbs (M2+M4) against BoNT/B1 intoxication: Changes of botulism symptoms in each mouse in the period of observation (2 weeks)” (line 572).

Figure S1

Therapeutic and preventive effects of HuMAbs (M2+M4) against BoNT/B1 intoxication: Changes of botulism symptoms in each mouse in the period of observation (2 weeks).

(A) Post-exposure treatment mouse model (correlated with Figure 5A). Mice were orally administered progenitor toxin (L-PTC, 10 ng), and subsequently administered M2+M4 (0.5 µg each) by i.p. injection at 12, 24, or 36 h after the oral administration of L-PTC. n = 5 per group. (B) Pre-exposure prevention mouse model (correlated with Figure 5B). Mice received i.p. injection of M2+M4 (0.5 µg each) and were then challenged 1, 3, 5, or 7 days later with i.p. administration of 10 i.p. LD50 BoNT/B1. Cnt, n = 7 per group, M2+M4 (1 day), n = 5 per group, M2+M4 (3 days), n = 5 per group, M2+M4 (5 days), n = 5 per group. M2+M4 (7 days), n = 5 per group. Scores were recorded for 2 weeks with symptoms ranging from “no symptoms” to scores of “mild”, “moderate”, and “severe”, defined by an increasing extent of botulism symptoms (fuzzy hair, muscle weakness, limb weakness, and respiratory failure).

9) Fig.5: the authors should modify the graphs. It is clear from the figure legend but it is not clear in the graph that the hours (5A) or days indicated (5B) refer to the BoNT/B injection or HuMAbs administration.

Response:

According to the reviewer’s suggestion, we revised Fig. 5. Thank you. 

10) In vitro data, using neurons, showing which step of botulinum neurotoxin mechanism of action is inhibited by the different antibodies will increase the novelty of the entire paper.  

Response:

The reviewer raised an important point. We are analyzing the neutralization mechanisms of M2 and M4 using neuronal cells, such as PC12 cells. However, we consider that many experiments (cell binding assay, translocation assay, VAMP2 cleavage assay, analysis of intracellular localization, etc.) will be needed to clarify the mechanism. It seems difficult to include these results in this paper. We added: “We are planning to reveal the step of the botulinum neurotoxin mechanism of action that is inhibited by M2 and M4 using neuronal cells, such as PC12 cells” (line 321).

Reviewer 2 Report

Line 11: The phrase “approximately ≥ 10 i.p. LD50/100 ug of antibody” and repeated in the text seems to redundant, since the ‘greater than and equal to’ symbols (≥) represent an approximation, so, writing the word “approximately” is indeed redundant.

Line 14: The term “80 i.p. LD50/2.5µg of antibodies” is not clearly understood.

Line Re line 80-81: The background readings for the non-vaccinated volunteers were mentioned but was not showed in the table.

Line 91:  The abbreviation ‘PBMCs’ was used before the words for the abbreviation were mentioned.

Line 94-95; Please explain the statement that; “These well were subjected to cell cloning by limited dilution”.

Line 97: The verb ‘were’ is not necessary in the phrase “—M1, M2, M4 and S1 were produced IgG—“

Lines251: Please correct the sentence that contains the words “at least our own model” to “at least in/for our own model”.

Lines 317—“Materials and Method”. The section should be written in more details, that allow more clarity, for example, in line 327-328 the statement “-----botulinum serotype B1 strain Okra was cultured using a cellophane tube procedure, and the culture supernatant was obtained”. Such should be written in more detail that offer more clarity.

Author Response

Dear Reviewer 2,

Re.: “Fully human monoclonal antibodies effectively neutralizing botulinum neurotoxin serotype B” by Takuhiro Matsumura, Sho Amatsu, Ryo Misaki, Masahiro Yutani, Anariwa Du, Tomoko Kohda, Kazuhito Fujiyama, Kazuyoshi Ikuta, and Yukako Fujinaga .

We are grateful for the helpful comments from you on the original version of our manuscript. We have addressed all of the comments as described below, and we hope that the explanations and revisions of our work are satisfactory.

Line 11: The phrase “approximately ≥ 10 i.p. LD50/100 ug of antibody” and repeated in the text seems to redundant, since the ‘greater than and equal to’ symbols (≥) represent an approximation, so, writing the word “approximately” is indeed redundant.

Response:

This has been revised. Thank you.

Line 14: The term “80 i.p. LD50/2.5µg of antibodies” is not clearly understood.

Response:

“80 i.p. LD50/2.5 µg of antibodies” means M2 + M4 (1.25 + 1.25 µg) neutralized the 80 LD50 BoNT/B1. We revised the sentence: “The combination of two HuMAbs, M2 and M4, was able to completely neutralize BoNT/B1 with a potency greater than 80 i.p. LD50/2.5 µg of antibodies,” to “The combination of two HuMAbs, M2 (1.25 µg) and M4 (1.25 µg), was able to completely neutralize BoNT/B1(80 i.p. LD50) with a potency greater than 80 i.p. LD50/2.5 µg of antibodies” (line 12).

Line Re line 80-81: The background readings for the non-vaccinated volunteers were mentioned but was not showed in the table.

Response:

“negative control plasma” (line 117) means plasma obtained from non-vaccinated volunteers. ELISA titers are expressed as the highest dilution factor with an absorbance at least twice that of plasma from non-vaccinated volunteers (negative control plasma) in Table 1. We added “obtained from non-vaccinated volunteers” (line 117).

Line 91:  The abbreviation ‘PBMCs’ was used before the words for the abbreviation were mentioned.

Response:

We added “peripheral blood mononuclear cells” (line 119) and deleted “peripheral blood mononuclear cells” (line 497).

Line 94-95; Please explain the statement that; “These well were subjected to cell cloning by limited dilution”.

Response:

We added details about “cell cloning by limited dilution” to Materials and Methods. We added: “Hybridomas were collected, diluted in medium, and then used to seed new plates (1–2 cells/well) to obtain a monoclonal cell population. The plates were incubated for 10–14 days.” (line 505).

Line 97: The verb ‘were’ is not necessary in the phrase “—M1, M2, M4 and S1 were produced IgG—“

Response:

This has been revised. Thank you.

Lines251: Please correct the sentence that contains the words “at least our own model” to “at least in/for our own model”.

Response:

This has been revised. Thank you.

Lines 317—“Materials and Method”. The section should be written in more details, that allow more clarity, for example, in line 327-328 the statement “-----botulinum serotype B1 strain Okra was cultured using a cellophane tube procedure, and the culture supernatant was obtained”. Such should be written in more detail that offer more clarity.

Response:

We revised the section of “5.2. Preparation of BoNT/A and BoNT/B” (line 441—) as follows:

  1. C. botulinum serotype B1 strain Okra was cultured using a cellophane tube procedure, and the culture supernatant was obtained. The culture supernatant was concentrated by 60% ammonium sulfate precipitation. After dialysis, progenitor toxins (M-PTC and L-PTC) were purified from the culture supernatant using an SP-Toyopearl 650M column (Tosoh, Tokyo, Japan) and a lactose gel column (EY Laboratories, San Mateo, CA, USA). and BoNT/B1 and a non-toxic component were prepared from L-PTC using the lactose gel column under alkaline conditions of pH 8.0, as described previously [64].

We added: “After cell cloning, cultures of BoNT/A1 or BoNT/B1-specific antibody producing-hybridomas were scaled up and we made cell stocks” (line 508).

We added: “The concentration of IgG was determined by BCA Protein Assay Kit (Thermo Fisher Scientific)” (line 514).

We added: “The concentration of recombinant protein was determined by BCA Protein Assay Kit” (line 525).

We added: “10 mM sodium phosphate buffer (pH 6.0) containing 0.1% gelatin was used for sample dilution.” (line 561)

Reviewer 3 Report

In the manuscript entitled “Fully Human Monoclonal Antibodies Effectively Neutralizing Botulinum Neurotoxin Serotype B” the authors present the results of generation of fully human monoclonal antibodies against botulinum neurotoxin serotype B using a murine–human chimaera fusion partner cell line. They showed that M2, which specifically binds to the light chain of botulinum neurotoxin serotype B1, neutralized its activity in a mouse bioassay, whereas M4, which binds to the C-terminal of a heavy chain, exhibit partial protection. However, the combination of these two was able to neutralize botulinum neurotoxin serotype B1 with a potency greater than 80 i.p. LD50/2.5 μg of antibodies. Moreover, this combination showed broad neutralization activity against other B subtypes, such as B2 and B6.

The manuscript is interesting and the study was planned well. They concluded that the combination of M2 and M4 is promising for therapeutics for botulinum neurotoxin serotype B intoxication, which makes the manuscript complementary. Taking these to the account, I strongly suggest to publish it without any further corrections.

Author Response

Thank you so much for your positive feedback.

Round 2

Reviewer 1 Report

The authors fully adressed my concerns. 

Author Response

Thank you so much for your helpful comments.